# OpenReview forum: "WildTeaming at Scale: From In-the-Wild Jailbreaks to (Adversarially) Safer Language Models"
_NeurIPS.cc/2024/Conference — NeurIPS 2024 poster_

### Official Review · Reviewer_wDdc · 2024-07-04

**Soundness:** 3
**Presentation:** 3
**Contribution:** 4
**Rating:** 6
**Confidence:** 4

**Summary:**

This paper introduces an automatic red-teaming framework that mines in-the-wild user chat logs and discovers various unique clusters of novel jailbreak tactics, which results in more diverse and successful adversarial attacks compared to state-of-the-art jailbreaking methods. Based on this framework, this paper creates a large-scale open-source synthetic safety dataset with much vanilla and adversarial prompt-response data pairs. This dataset enables the machine learning communities to examine the scaling effects of data and the interplay of data properties.

**Strengths:**

1. This paper conducted a comprehensive study on the current in-the-wild jailbreak attacks. The two challenges mentioned in the Introduction are well addressed in this paper.

2. The two-stage framework is reasonable. The first step involves collecting the real-world user conversation logs and deconstructing them, while the second step composes the jailbreak tactics and construct them with vanilla harmful queries.

3. The experiment are thorough and demonstrate the efficiency of the framework. The experimental results in Section 5.2 validate the utility of the benchmark in safety training.

**Weaknesses:**

1. Some method details missed. This paper lacks details on how the tactics are selected for constructing the final jailbreak prompt from an initial harmful prompt, as mentioned in lines 126-128. Table 1 lists many tactics, but it is unclear how these tactics are chosen.

2. Some metrics are vague: The process for constructing jailbreak prompts, as described in Sections 2 and 3.1, is fixed. However, the metric “Query” in Table 2 changes with the target model. This discrepancy needs explanation. Are there additional details about the method not mentioned in the paper?

3. The evaluation methods is not rationale enough: In Table 2, diversity is evaluated using sentence embedding similarity. This straightforward method might not work in all cases. For example, if the similarity between A and B, B and C, C and A is 0.8, 0.2, and 0.3, respectively, the average similarity for A, B, C would be 0.55, 0.5, and 0.25, respectively, categorizing them into three parts since they are all lower than 0.75. However, they should be categorized into two parts {A+B, C}. Additional details and justification for the similarity-based metric are needed to prove its rationality.

**Questions:**

1. See (1) in weaknesses. How these tactics are selected? Are all these tactics available for almost all the harmful prompts? If not, how should you ensure the semantic rationality of the constructed jailbreak prompt?

2. See (2)&(3) in weaknesses. Could you give more details about each metric and the similarity-based metric and prove their rationality?

**Limitations:**

Authors have not demonstrated the limitations specifically. Limitations from this reviewer can be seen in Weaknesses and Questions.

---

> ### Author Rebuttal · Authors · 2024-08-07
>
> We greatly appreciate reviewer `wDdc` for their recognition of our comprehensive study and thorough experiments and for their support in accepting the paper. We will address their questions in the following section and are happy to follow up during the discussion period for any further inquiries.
>
> > Clarification of Method Details
>
> Thank you for raising the question about the method details. Due to space limitations during the initial submission, we provided a high-level description of the method in the main paper and detailed components of WildTeaming, including the attack model, target models, off-topic and low-risk pruners, and attack selectors, in Appendix §B.1. With more space allowed in the final paper, we will include more detailed method descriptions in the main paper to enhance readability. We address the specific question of **How are the tactics selected to compose attacks?** below.
>
> Table 1 shows that we identified 105K items and 5.7K unique clusters of novel jailbreak tactics from in-the-wild user data. Our goal is to use these diverse tactics to create effective adversarial attacks and reveal new model vulnerabilities. Therefore, we consider all tactics to expand the exploration space for jailbreak tactic combinations.
>
> Specifically, given a vanilla harmful query for the jailbreak experiments in Sec 3, our goal is to select N tactics from the entire pool of mined jailbreak tactics and apply them to transform the vanilla query into an adversarial attack. We used a prioritized random sampling strategy, selecting N/2 tactics from the common pool (tactics occurring ≥3 times during mining) and N/2 from the less common pool (tactics occurring <3 times). Note that there are many more uncommon tactics than common tactics, and therefore, sampling N/2 separately from each pool will result in more common tactics being selected than uniform random sampling across the entire pool. We chose this strategy to prioritize the common tactics more to mimic how users use them more frequently. Another experimental design choice we made is that since the PAIR baseline explicitly uses the tactic of “adding seed leading sentence” in their attack revision prompt, to be consistent with their setup, we also fixed this tactic in our tactics selection choices and selected the remaining (N-1) tactics following the prioritized random sampling strategy described above. Regarding the choice of N, we run a series of ablations to test the optimal N and include results in Table 19 of the Appendix. Our results show that N=4 leads to the best performance on both standard and diversity metrics, so we adopt this setup for all the main experiments.
>
> To ensure the semantic rationality of the constructed jailbreak prompt, we acknowledge that not all combinations of the sampled tactics choices will be fully compatible, e.g., we may randomly select tactics at odds with each other. To mitigate the possibility of introducing low-quality attacks generated with conflicting tactics, during the attack generation process, we specifically instruct the attacker model to maximize the usage of all selected tactics but ignore incompatible ones if necessary to keep the end attack “logically coherent and fluent” as much as possible. However, even deliberate instructing models may still result in suboptimal attacks. To ensure the quality of the final attack candidate, we further apply low-risk and off-topic pruners to eliminate attack candidates that do not precisely portray the original harmful intent.
>
> **We will refine our method descriptions in the final paper, ensuring all necessary and additional details are included for clarity.**
>
> > Clarification of Evaluation Metrics
>
> Thank you for pointing out the need for clarity in our evaluation metrics descriptions. We have included detailed descriptions of all standard and diversity evaluation metrics in §3.2 (L145-161) and will ensure they are polished in the final paper for clear understanding. Below, we clarify two specific metrics in Table 2 that arose during reviews.
>
> - **Why does “Query” change across target models?**
>
> As mentioned in L148-149, “Query” was introduced to measure attack efficiency, representing “the number of queries needed to reach a successful attack.” Generation-based jailbreak methods, like WildTeaming and PAIR, employ a “trial-and-error” approach, repeatedly proposing candidate attacks and testing them against the target model until a promising attack is identified. “Query” denotes the number of these repetitions required to achieve a successful attack, thus indicating the speed at which a jailbreaking method can successfully attack. Thus, for different target models with different levels of safeguard, a method will take a different number of trials to succeed. In the final paper, we will change this metric to “Number of Attack Trials” to better capture its meaning.
>
> - **How do we use sentence embedding similarity for measuring“Diversity”?**
>
> We define a new suite of diversity metrics to assess the ability of jailbreak methods to identify *multiple unique successful attacks*. Specifically, to decide if an attack $a_{\text{new}}$ is unique compared to a pool of existing attacks $A = $ {$a_1, a_2, …, a_n$}, we compute the sentence embedding similarity between $a_{\text{new}}$ to all $a_i$ in $A$, i.e., cos_sim($E$[$a_{\text{new}}$], $E$[$a_i$]), where $E$ indicates sentence embedding. We consider $a_{\text{new}}$ to be a unique new attack only if it’s *dissimilar* to $\forall a_i \in A$, i.e., cos_sim($E$[$a_{\text{new}}$], $E$[$a_i$]) $< 0.75$, $\forall a_i \in A$. If $a_{\text{new}}$ shows similarly to *any* $a_i \in A$, it’ll be considered redundant. We do not average the sentence embedding score for $a_{\text{new}}$ to decide its uniqueness as interpreted by the reviewer, and **we will resolve this ambiguity in our final paper**.

---

> > ### Comment · Reviewer_wDdc · 2024-08-11
> >
> > Thanks for the response. The response addresses most of my concern.

---

> ### Author Response · Authors · 2024-08-12
> **Thank you for taking the time to review our rebuttal and for recognizing that it helps address previous concerns!**
>
> Dear Reviewer `wDdc`,
>
> Thank you for taking the time to review our rebuttal and for recognizing that it helps address previous concerns! If our response has satisfactorily resolved the questions raised, we would greatly appreciate your consideration of adjusting the score, if deemed appropriate.
>
> Thank you once again for your time and effort.
>
> Sincerely,
> Authors of Paper 12439

---

### Official Review · Reviewer_idQi · 2024-07-05

**Soundness:** 3
**Presentation:** 2
**Contribution:** 3
**Rating:** 5
**Confidence:** 5

**Summary:**

The paper introduces WILDTEAMING, an automatic red-teaming framework that mines user-chatbot interactions to discover novel jailbreak tactics against large language models (LLMs). It uncovers vulnerabilities of frontier LLMs, leading to more diverse and successful adversarial attacks. The authors also created WILDJAILBREAK, a large-scale open-source synthetic safety dataset comprising 262K prompt-response pairs to enhance safety training. The paper demonstrates the effects of data properties and model capabilities during safety training, identifies training properties for balanced safety behaviors, and shows that WILDJAILBREAK improves model safety without diminishing general capabilities. The authors emphasize the need for shared norms and technical standards in LLM safety and call for further research into safety training methods and comprehensive safety evaluation.

**Strengths:**

Originality:
- The WILDTEAMING approach of mining jailbreak tactics from real-world user interactions and composing them into diverse attacks is novel and creative.
- The creation of the WILDJAILBREAK dataset with four contrastive components (vanilla/adversarial harmful/benign queries) is an original contribution.
- The paper introduces new diversity metrics for evaluating jailbreaking methods.

Quality:
- The methodology appears rigorous, with careful mining of jailbreak tactics, automatic scaling using GPT-4, and extensive experiments.
- Comparisons to multiple strong baselines demonstrate the effectiveness of the approach.
- The analysis of scaling effects and interplay between data properties and model capabilities is thorough.

Clarity:
- The paper is generally well-structured and clearly written.
- Key concepts and methods are explained in detail.
- Figures and tables effectively illustrate the approach and results.

Significance:
- The WILDTEAMING framework provides a practical way to identify diverse vulnerabilities in language models at scale.
- The WILDJAILBREAK dataset is a valuable open resource for safety training.
- The insights on scaling effects and balanced safety behaviors are important for developing safer AI systems.
- The work addresses critical challenges in AI safety and ethics.

Overall, this paper makes substantial original contributions, demonstrates high-quality research, clearly communicates the methods and findings, and has significant implications for improving the safety and robustness of large language models. The combination of novel techniques, extensive experimentation, and practical resources make this a strong and impactful work in the field of AI safety.

**Weaknesses:**

1. Limited analysis of ethical implications:
While the paper focuses on improving AI safety, it lacks a thorough discussion of the ethical implications of creating and releasing a large dataset of harmful queries and jailbreak tactics. The authors should address:
- Potential misuse of WILDJAILBREAK by malicious actors
- Safeguards implemented to prevent unintended harm
- Ethical considerations in mining real-world user interactions for jailbreak tactics

Suggestion: Include a dedicated section on ethical considerations, discussing the above points and outlining measures taken to ensure responsible use of the dataset and methods.

2. Insufficient comparison to human-generated jailbreaks:
The paper compares WILDTEAMING primarily to automated methods like PAIR and AutoDAN. However, it lacks a direct comparison to human-generated jailbreaks, which could provide valuable insights into the effectiveness of the approach.

Suggestion: Conduct a small-scale study comparing WILDTEAMING-generated jailbreaks to those created by human experts or crowdworkers. This would help contextualize the method's performance relative to human creativity.

3. Limited exploration of model-specific vulnerabilities:
While WILDTEAMING is presented as a model-agnostic approach, it's possible that certain models may be more vulnerable to specific types of jailbreak tactics. The paper doesn't explore this aspect in depth.

Suggestion: Analyze the effectiveness of different jailbreak tactic categories across various model architectures and sizes. This could reveal interesting patterns in model vulnerabilities and inform more targeted defense strategies.

4. Lack of long-term effectiveness assessment:
The paper doesn't address how the effectiveness of WILDTEAMING-generated jailbreaks might change over time as models are updated or as the approach becomes widely known.

Suggestion: Discuss the potential limitations of the approach in a "Limitations and Future Work" section. Consider proposing a longitudinal study to assess the long-term effectiveness of the generated jailbreaks.

5. Insufficient analysis of computational costs:
While the paper mentions that WILDTEAMING has "low computational costs," it doesn't provide a detailed analysis of the computational requirements compared to other methods.

Suggestion: Include a quantitative comparison of computational resources required for WILDTEAMING versus other jailbreaking methods. This would strengthen the claim of practicality for scaling up.

6. Limited discussion of failure cases:
The paper focuses primarily on the successes of WILDTEAMING but doesn't provide an in-depth analysis of cases where it fails or performs suboptimally.

Suggestion: Include a section analyzing failure cases, discussing the types of jailbreaks that WILDTEAMING struggles to generate or identify. This could provide valuable insights for further improvement of the method.

7. Lack of ablation studies on jailbreak tactic categories:
While the paper presents an ablation study on pruning methods, it doesn't explore the relative importance of different categories of jailbreak tactics.

Suggestion: Conduct ablation studies removing different categories of jailbreak tactics to understand their individual contributions to the overall effectiveness of WILDTEAMING.

**Questions:**

1. Ethical considerations and safeguards:
   - What specific measures have been taken to prevent misuse of the WILDJAILBREAK dataset?
   - How do you address potential privacy concerns related to mining real-world user interactions?
   - Have you consulted with ethics experts or an ethics review board during this research?

2. Comparison to human-generated jailbreaks:
   - Have you considered conducting a comparative study between WILDTEAMING and human experts in generating jailbreaks?
   - If so, what challenges do you foresee in such a comparison?

3. Model-specific vulnerabilities:
   - Did you observe any significant differences in the effectiveness of certain jailbreak tactics across different model architectures or sizes?
   - Could you provide more detailed analysis on how WILDTEAMING performs across a wider range of models?

4. Long-term effectiveness:
   - How do you anticipate the effectiveness of WILDTEAMING-generated jailbreaks to change over time as models are updated?
   - Have you considered strategies for maintaining the relevance of the approach as AI systems evolve?

5. Computational efficiency:
   - Can you provide a more detailed comparison of the computational resources required for WILDTEAMING versus other jailbreaking methods?
   - How does the efficiency of WILDTEAMING scale with the size of the target model?

6. Analysis of failure cases:
   - Could you elaborate on specific types of jailbreaks or scenarios where WILDTEAMING performs suboptimally?
   - How might these failure cases inform future improvements to the method?

7. Jailbreak tactic categories:
   - Have you conducted ablation studies on different categories of jailbreak tactics?
   - If so, what insights did you gain about the relative importance of various tactic types?

8. Generalizability beyond chatbots:
   - How well do you expect WILDTEAMING to generalize to other types of language models or AI systems beyond chatbots?
   - Have you tested or considered applying the method to other domains?

9. Integration with existing safety measures:
   - How do you envision WILDTEAMING and WILDJAILBREAK being integrated into existing AI safety practices?
   - Have you explored combining your approach with other safety techniques?

10. Potential for defensive applications:
    - Could the WILDTEAMING framework be adapted for proactive defense against jailbreaks?
    - Have you considered using the method to automatically generate safety training data in real-time?

11. Scalability of the approach:
    - As language models continue to grow in size and capability, how do you expect the scalability and effectiveness of WILDTEAMING to be impacted?
    - Are there any foreseeable limitations to the approach as AI systems become more advanced?

**Limitations:**

1. Ethical concerns: The creation and release of a large dataset of harmful queries and jailbreak tactics could potentially be misused by malicious actors.

2. Generalizability: The effectiveness of WILDTEAMING may be limited to the types of models and interactions present in the mined datasets (LMSYS-1M and WILDCHAT).

3. Temporal relevance: As AI models evolve, the jailbreak tactics discovered by WILDTEAMING may become less effective over time.

4. Dependency on existing language models: The COMPOSE stage relies on off-the-shelf LLMs like Mixtral-8×7B and GPT-4, which may introduce biases or limitations.

5. Lack of human baseline: The paper doesn't provide a direct comparison between WILDTEAMING and human-generated jailbreaks, which could offer valuable context.

6. Model-specific vulnerabilities: The approach may not fully capture vulnerabilities that are specific to certain model architectures or training paradigms.

7. Computational requirements: While mentioned as having "low computational costs," the exact resource requirements and scalability of WILDTEAMING are not thoroughly detailed.

8. Limited exploration of failure cases: The paper focuses more on successes and less on scenarios where WILDTEAMING is less effective.

9. Potential for adversarial adaptation: As jailbreaking methods become known, AI developers may adapt their safety measures, potentially reducing the long-term effectiveness of WILDTEAMING.

10. Privacy considerations: Mining real-world user interactions for jailbreak tactics may raise privacy concerns, even if the data is anonymized.

11. Scope limitation: The focus is primarily on text-based language models and may not generalize well to other types of AI systems.

12. Potential for over-optimization: There's a risk that models trained on WILDJAILBREAK might become overly cautious, potentially affecting their performance on benign tasks.

13. Dependency on existing safety classifiers: The method relies on existing tools like the OpenAI Moderation API and LLAMA-GUARD, which may have their own limitations or biases.

14. Limited cross-cultural analysis: The paper doesn't extensively explore how jailbreak tactics might vary across different languages or cultural contexts.

15. Potential for false positives: The method might incorrectly classify some benign queries as jailbreak attempts, leading to unnecessary restrictions.

---

> ### Author Rebuttal · Authors · 2024-08-07
>
> We greatly appreciate reviewer `idQi`'s thorough, insightful questions & constructive suggestions. We're delighted by the recognition of our work's originality, quality, clarity, & significance. **In general response we answer 4 shared questions, and respond to other questions below.** Our responses will be concise here (limit is 6K characters) but we're happy to follow-up in discussions & reflect full changes in the final paper.
>
> > Model Specific Vulnerabilities
>
> We had a model-specific analysis in Table 27 of Appendix §C.3, which we convert into a heat map for clearer insights (see Table (a) in rebuttal PDF). **We will include this discussion in the final paper.** Here's a brief summary:
>
> - Larger models within the same family are not always more robust, e.g., larger Gemma1/1.1 & Vicuna show lower ASR, but Tulu2 & Llama3 have opposite trend.
>
> - For GPTs, newer versions aren't always more robust than older ones, e.g., the 0125 GPT-3.5/4 checkpoints are less robust than their predecessors, suggesting different checkpoints might prioritize different capabilities, leading to occasional safety trade-offs.
>
> - Some tactics have a 0 ASR for certain models, indicating that specific jailbreak types may be patched during model training, leading to low ASR for those types but not necessarily for all types.
>
> > Jailbreak Tactic Choices Ablations
>
> Table 19 in the Appendix includes additional ablations on tactic choices. We will provide clearer references to these results in the final paper. Below summarizes key findings.
>
> First, we studied the effect of the number of selected tactics on attack composition. Sampling 4 tactics resulted in the highest ASR with high attack efficiency.
>
> | # Tactics | $ASR_{30}^{5} \uparrow$ | $Query_{30}^{5} \downarrow$  |
> |-|-|-|
> | 1 | 65.4              | 11.89 |
> | 2 | 74.6              | 10.73 |
> | 3 | 77.1              | 9.92  |
> | 4 | **80.5**         | **9.94** |
> | 5 | 78.0              | 11.43 |
> | 6 | 79.0              | 10.16 |
>
> Given the large number of mined jailbreak tactics (105K items in 5.7K clusters), exhaustive ablations on each tactic type are computationally intractable. We focused on one specific tactic: “adding a lead sentence,” used in the PAIR baseline's attack prompt. To align with their setup, we ablate by fixing this tactic while randomly sampling others. Table 19 shows that fixing this tactic and sampling 3 others yielded the best scores ($ASR_{30}^{5}$: 86.8), so we adopt this setup for final evaluations.
>
> Finally, Figure (a) of the rebuttal PDF reveals more insights of tactics choices. Compared to the overall average of 48.1, tactics like "fictional scenarios" (57.1) and "enforcing rule-breaking" (53.2) lead to higher ASR. While "adding distractor content" has a lower ASR (39.4), "excessive" distractions increase it to 48, showing the impact of "information overload." Syntactic obfuscations (53.5) are more effective than style and framing obfuscations (42.6). **We will discuss these in the final paper.**
>
> > Human Comparisons
>
> We appreciate the reviewer's suggestion of comparison to human-written jailbreaks. While this is a valuable angle, direct comparisons with WildTeaming pose challenges and unclear implications. We will detail our explanation below and add this to future work in our final paper.
>
> Early human red-teaming efforts quickly became impractical due to prohibitive costs & lack of scalability. Expert red-teamers require deep knowledge of modern LMs & AI safety risks, making their training challenging & expensive, even for well-funded companies. Even so, past human red-teaming data often includes simple & noisy queries. These limitations led to the initial development of automatic methods, which WildTeaming further advances their scalability & practicality (see general response). The comparison of human- & auto-attacks depends on variables like the attacker's expertise, prior knowledge of attack types, modes of human-model interactions, beyond the current research scope. We will discuss it in future work.
>
> Regardless, **our paper included comparisons involving previous human-written red-teaming datasets**. In Table 4 & 33, we compare models trained with previous human data (HH-RLHF, SafeRLHF, DAN) to model trained with WildJailbreak. Our model substantially outperforms those baselines, e.g., 3.1 vs. 66.0 on HarmBench, among many other tasks.
>
> > Others
>
> Due to space limits, we provide brief answers for points below but happy to elaborate during discussions.
>
> - **Over-Refusals:** We share the concern about excessive refusals in benign cases, and include extensive benign contrastive data in WildJailbreak, designed to reduce over-refusals (see §4.1). Our ablations show that excluding this data leads to significant over-refusals, while its inclusion minimizes the rate (see Table 4).
>
> - **Failure Cases:** We will include a dedicated sub-section in the final paper to detail failure cases and inform the next iteration of method design.
>
> - **Cross-Cultural Analysis:** We recognize the importance cross-language and cultural analysis. This work focuses on English user interactions. We will address this in limitations and future work.
>
> - **Integration into Existing Safety Practices:** Our open-sourced safety training data will lead to models with improved intrinsic safeguards captured by model parameters. External defenses, such as filtering with safety classifiers & decoding, can be seamlessly integrated with WildJailbreak-trained models for greater robustness.
>
> - **Proactive Defense:** Applying WildTeaming real-time and for proactive defense is feasible, involving an iterative process where WildTeaming is in the loop. We will discuss this in future work.
>
> - **Beyond Language Models:** Addressing the significant unresolved challenge of enhancing LM safety is a major contribution. Thus, we respectfully disagree that the modality constraint is a limitation. We will discuss future works that apply WildTeaming in areas like vision-language models.

---

### Official Review · Reviewer_ZrdC · 2024-07-09

**Soundness:** 3
**Presentation:** 2
**Contribution:** 3
**Rating:** 7
**Confidence:** 3

**Summary:**

This work proposes, for LLM safety, a new jailbreak attack method, WildTeaming, and a new safety alignment dataset, WildJailbreak. WildTeaming first manually identify seed jailbreak tactics from large-scale in-the-wild adversarial user query datasets and then compose jailbreak attack by rewriting a vanilla harmful query into an adversarial harmful query based on the multiple identified tactics using a LLM. WildJailbreak is then built based on WildTeaming with balanced distribution of vanilla and adversarial, harmful and benign, samples.

**Strengths:**

1. The scale and the diversity of attack and safety alignment dataset are large. They can thus complement the existing works.
2. The effort of identifying comprehensive tactics for jailbreak is appreciated.
3. The idea of using a LLM to compose multiple jailbreak tactics into one jailbreak attack prompt is interesting.
4. The opensource of large-scale safety alignment dataset could contribute a lot to future safety alignment research.

**Weaknesses:**

1. The discussion of related works is insufficient. Specifically, many jailbreak methods are only referred yet without any description at Line 301. There are many different categories of jailbreak attack methods. They should be disccused in more details. The same issue is also applied to the review of safety enhancement methods of LLMs. Besides, the paper can be enhanced by providing more detailed comparison with the closely related works.
2. The motivation of diverse jailbreak tactics is weak. Following the convention of adversarial ML, given a vanilla harmful query, the objective should be to find the most effective tactic to rewrite the query to trigger the harmful output, instead of to find many different tactics. Therefore, I am not convinced by the evaluation protocol of comparing different jailbreak methods in terms of diversity. Furthermore, it seems that increasing the diversity of jailbreak tactics seems to only result in a marginal improvement in ASR or even lower than PAIR in Tab. 2 for Vicuna-7B.
3. The evaluation and comparison of the proposed methods are insufficient. First, the proposed attack is not tested against a broad range of defenses, some alternatives including rephrasing [1] and prompt engineering [2]. Second, the proposed safety alignment dataset is not compared against other external methods. Safety-Tuned LLaMAs is a closely related work but not empirically compared at all. Therefore, I agree with the effectiveness of proposed methods but feel hard to compare them against the latest progress in the field.

**Questions:**

1. One of the listed tactics in Tab. 1 (right) is "ignore prev. instructions". Is not it a method of prompt injection?

**Limitations:**

The limitations and ethical considerations of the proposed method are discussed in section 7.

---

> ### Author Rebuttal · Authors · 2024-08-07
>
> We greatly appreciate reviewer `ZrdC` for recognizing our work’s unique open-source contribution with “scale and diversity”, our “effort of identifying comprehensive tactics for jailbreak”, and our method being “interesting”. We address their insightful questions in the following section and are happy to follow up during the discussion period for any further inquiries.
>
> > Related Works
>
> Due to the space limit, we kept related work brief in the initial submission. We have already enhanced the discussions on both **Red-Teaming and Jailbreaking LLMs** and **Safety Evaluation and Training of LLMs**, which will be reflected in the final paper. If the reviewer is interested, we would be happy to share the updated version *during the discussion period*, as there is a strict word limit for this initial rebuttal.
>
> > Importance of Diversity
>
> We want to clarify that the core motivation of our work is **to build safe language models that are robust against a wide range of vanilla and adversarial attacks**, rather than *developing an attack method that targets a specific type of attack with a high success rate*. **Diversity** is crucial for evaluating the **practicality** of a red-teaming method in its ability to (1) broadly reveal model vulnerabilities and (2) synthesize systematic data to enhance model safety.
>
> -  #### Broadly revealing model vulnerabilities:
>
> The ultimate goal of automatic red-teaming methods is to **broadly** reveal model vulnerabilities, allowing researchers to **systematically** improve safeguards. Yet, the focus on single-attack effectiveness bias many methods to identifying a narrow range of vulnerabilities, as shown by the low diversity scores in Tables 2 and 18 of Appendix §B.4. For example, AutoDAN generates only one type of attack even after 30 attempts, making it ineffective for uncovering broad model vulnerabilities.
>
> For a red-teaming method to be practically useful, it must **efficiently identify diverse model vulnerabilities for scalable red-teaming**. Without considering attack diversity, methods may over-optimize for a single successful attack and fail to identify different attacks, reducing their practicality for broad red-teaming. Thus, we define new diversity metrics to demonstrate WildTeaming's advantage in identifying diverse successful attacks.
>
> -  #### Synthesizing systematic data to enhance model safety:
>
> Improving model safeguards is more crucial than merely revealing vulnerabilities. Training models with safety alignment data enhances their safety, but existing open data is scarce and often low quality. Our paper's Table 4 demonstrates that the Tulu2 model, trained with existing safety data, performs poorly on downstream evaluation tasks.
>
> Thus, we aimed to create effective open-source synthetic safety training data with **large scale** and **diverse coverage** to support open science in AI safety research. Like revealing model vulnerabilities, slow red-teaming methods producing narrow types of attacks are inadequate for creating large-scale, high-quality data. Thus, we developed WildTeaming, which generates diverse attacks by a combinatorial application of diverse tactics, enabling scalable safety data creation.
>
> **Overall, WildTeaming achieves up to 4.6x higher diversity scores compared to other baselines across various models, highlighting its superiority in identifying diverse vulnerabilities and creating extensive safety data.** While WildTeaming may have a slightly lower standard ASR for Vicuna-7B, it generally achieves comparable or better ASR across models. The substantial improvement in diversity metrics (up to 28.6% absolute gain) outweighs the occasional slight decrease in standard ASR scores (around 1%).
>
> > Extended Comparisons
>
> We thank the reviewer's suggestions for additional comparisons to demonstrate our method's advancement. Specifically, the reviewer proposed: (1) testing our attacks against defense methods like prompt engineering, and (2) comparing our safety alignment dataset with existing methods, especially Safety-Tuned LLaMAs (STL).
>
> We fully support comparison with models trained on existing datasets. In Section 5, we present baseline models trained on previous safety training resources (HH-RLHF, SafeRLHF, DAN). As shown in Table 6, this baseline (T2M-public-safety) underperforms in downstream evaluations compared to the model trained with WildJailbreak (e.g., 66.0 vs. 3.1 on HarmBench; lower is better). Similar performance gaps are observed across benchmarks, and ablations with different dataset combos confirm these findings (see Table 34 in Appendix §D.4). **These results provide solid evidence that WildJailbreak is more effective than existing safety alignment datasets.**
>
> Initially, we excluded Safety-Tuned LLaMAs (STL) from our baseline due to its small size (2.5K vs. 208K) and prompts derived from HH-RLHF. During the rebuttal period, we tested this baseline. Surprisingly, despite its small size, STL outperforms other baselines with existing data (e.g., 32.7 vs. 66.0 on HarmBench), likely due to their higher-quality model responses. Nonetheless, our model outperforms STL **substantially** (e.g., 3.1 vs. 32.7 on HarmBench, 14.0 vs. 59.3 on DAN, 1.7 vs. 72.0 on WildJailbreak, lower is better). These results again underscore the value of WildJailbreak. **We will include STL comparisons in the final paper.**
>
> Regarding point (1), we respectfully push back that testing our attacks against those mentioned defenses is mandatory. While these external defenses can improve protection, they do not fundamentally alter model parameters or enhance intrinsic robustness. Our research focuses on improving intrinsic model safety via robust adversarial training. Our empirical results strongly support this agenda, regardless of testing against these defenses. We will discuss these defenses as complementary to adversarial training in future work.
>
> > "ignore prev. instructions"
>
> This tactic can also be used in jailbreak beyond prompt injection.

---

> > ### Comment · Reviewer_ZrdC · 2024-08-10
> >
> > Thanks much for the authors' thorough responses. Some of my concerns have been addressed. Kindly please find blow the remaining concerns:
> >
> > 1. Yes, I am interested. I would appreciate it if the authors could share the updated version.
> > 2. I agree with the authors on the importance of diverse attacks in enhancing safety training. However, from the perspective of attacking, I still feel the point of "broadly revealing vulnerabilities" is weak. The proposed diversity lies in the different strategies of  implementing the same harmful capability, instead of implementing different harmful capabilities. For example, to make the model teach how to make a bomb, malicious users seek for the most effective, instead of many, strategy to achieve the purpose. The only point of diversity in attacking is to bypass some over-optimized (for a subset of attacks) defense mechanism. However, given the observed marginal increase in ASR in tab. 2, it seems that the aforementioned point is not a major concern.
> > 3. Thanks for providing the results of STL which convinces me the effectiveness of proposed safety alignment dataset. I also personally believe that learning an intrinsically safe model is promising and necessary for safe advanced AI. However, I don't think this means the other defense approaches like SmoothLLM [1] and Self-Reminders [2] (sorry for the missing citations in the initial review) should not be used and thus not attacked against. In fact, there are some arguments that safe LLM should be approached in a systematic way, e.g., the Llama Prompt Guard [3]. Therefore, I think the evaluation against these alternative defense methods is necessary to comprehensively demonstrate the effectiveness of proposed jailbreak attack method.
> >
> > [1] Robey et al., SmoothLLM: Defending Large Language Models Against Jailbreaking Attacks, 2023
> >
> > [2] Xie et al., Defending ChatGPT against jailbreak attack via self-reminders, Nature Machine Intelligence, 2023
> >
> > [3] https://ai.meta.com/blog/meta-llama-3-1-ai-responsibility/

---

> ### Author Response · Authors · 2024-08-12
> **Thank you for taking the time to review our rebuttal! Follow-up Response (post 1)**
>
> We sincerely appreciate reviewer `ZrdC`'s follow-up questions and active engagement during the discussion! In this response, we address the additional inquiries and strive to fully resolve any remaining concerns. If our response has satisfactorily resolved the questions raised, we would greatly appreciate your consideration of adjusting the score, if deemed appropriate. We also welcome further discussion if the reviewer has additional points and are happy to incorporate any further suggestions into our final paper.
> > Full Draft of Updated Related Work
>
> - **Red-Teaming and Jailbreaking LLMs.**
>
> Early attempts at red-teaming and understanding LLM vulnerabilities have focused on hand-crafting prompts and analyzing model responses [2, 22, 54, 69]. However, manual methods had quickly become impractical due to their prohibitive costs and lack of scalability. Thus, automated red-teaming and jailbreaking methods are developed for more scalable audit of model vulnerabilities [56]. One genre of methods involves gradient optimization that requires back-propagating through model parameters [96, 27, 28, 67]. However, they are computationally expensive, cannot be applied to black-box models, and often result in gibberish texts. There are also generation-based approaches (most related to our work) which generate jailbreaking prompts directly or through iterative edits [8, 49, 43, 44, 56, 7, 52, 87, 40, 88, 90, 19]. Other jailbreaking works study attacks during decoding time (e.g., decoding configurations [32], logit manipulation [92]), in other modalities (e.g., vision-language [68, 85, 66], LLM agents [61]), under multilingual settings [20, 86, 60], in programming mode [41], or through multi-turn interactions [62, 83, 63]. However, most existing automatic red-teaming and jailbreak methods rarely result in large-scale training resources for model safety enhancement due to their limited coverage of attack strategies and risk types, slow speed, or closed-source access [65]. WildTeaming differs from previous works by efficiently composing *diverse* adversarial attacks utilizing real-world jailbreak tactics mined from in-the-wild user-chatbot interactions. WildTeaming allows scalable synthetic safety training data generation in addition to simply showing its attack efficacy.
>
> - **Safety Evaluation and Safety Training of LLMs.**
>
> Many red-teaming efforts on LLMs have been formalized as benchmarks for evaluating model vulnerabilities—these typically are composed of harmful prompts that models should refuse [6, 78, 77, 71, 51, 23, 75, 9]. Meanwhile, to mitigate the potential byproducts of safety training, other benchmarks measure exaggerated safety behavior on benign queries [64, 15]. While LLM safety evaluation has been an active area of research, studies and resources for safety training have been *limited*, especially in adversarial settings [22, 16, 84]. Most related to our work in this space are Safety-Tuned Llamas [3] and SafeRLHF [16], which primarily focus on *vanilla harmful* queries by releasing small-scale safety training datasets and large-scale pairwise preference datasets, respectively. WildTeaming distinguishes from these works by releasing higher quality (shown by our training ablation experiments) and larger scale sequential instruction-tuning data consisting of both *vanilla* and *adversarial* queries. WildJailbreak also uniquely contains large-scale contrastive *benign* queries, in addition to harmful ones, used for mitigating exaggerated safety behavior (i.e., over-refusal). Finally, synthetic data has been used for LLM safety [7, 58,15]. Most relevant to our work is Rainbow Teaming [65], which uses synthetic data to populate a grid of attack spaces based on the attack style and risk category. However, their data and code are not publicly available. Our work differs in automatically mining diverse human-devised jailbreak tactics rather than manually defining attack styles [65], creating a large-scale open safety training resource that supports extensive safety training experiments.
>
> *We have omitted the citations corresponding to each number entry to avoid cluttering the response. However, we are again happy to share them if this is helpful information for the reviewer.*

---

> ### Author Response · Authors · 2024-08-12
> **Follow-up Response (cont. post 2)**
>
> > Further Clarification on the Diversity Metrics
>
> We appreciate the reviewer's further comments on the diversity measure of attacks, which has helped us understand the source of our differing views. We believe this discrepancy arises from the distinct roles we attribute to automatic jailbreak and red-teaming methods. The reviewer assesses these methods from a *security attacking perspective*, where the focus is on successfully breaking a model, typically quantified by the success rate of a single attack. In this project, we view these methods from a *model developer's perspective*, treating them as scalable model auditing tools designed to automatically and comprehensively identify model vulnerabilities. For such auditing purposes, diversity is crucial because it allows the method to uncover a wide range of failures. This emphasis on diversity in red-teaming is also supported by previous work (e.g., Rainbow Teaming [1]), though their data and code are not publicly available for comparison. By defining diversity metrics in our study, we hope to encourage future methods to consider incorporating the practical aspect of better supporting model auditing. *We will clarify our motivation further in the final paper.*
>
> In response to the reviewer’s point regarding our method enhancing diversity in tactics for implementing the same type of harm, rather than exploring different harm categories, it is important from the model developer's standpoint to consider both dimensions: (1) harm categories (e.g., misinformation, cybersecurity) and (2) jailbreak tactics. Our objective is to identify as many instances of harmful model responses as possible, covering both diverse harm categories and various methods of eliciting harm within a single category. Thus, both dimensions are crucial for improving model safety, also noted by previous literature [1]. The standard evaluation setup for auto-jailbreak methods typically fixes the harm category dimension, focusing on applying jailbreak strategies to provoke a specific harmful response, without uncovering new types of harmful behaviors. Within this evaluation framework, our work contributes by employing diverse novel strategies to trigger the targeted harmful behavior, which may aid in developing robust defenses against such behavior despite the variety of elicitation techniques. Although we do not explicitly focus on diversifying harm categories during the attack experiments due to the constraints of the standard evaluation setup, we address this in the construction of the WildJailbreak dataset. This dataset includes 50K harmful queries across 4 high-level categories (e.g., malicious uses, harmful language, misinformation, privacy) and 12 subcategories, significantly expanding the resources available for defining diverse vanilla harmful behaviors (examples in Table 23 in the Appendix).
>
> [1] Rainbow teaming: Open-ended generation of diverse adversarial prompts, 2024.

---

> ### Author Response · Authors · 2024-08-12
> **Follow-up Response (cont. post 3)**
>
> > Further Comparison of Defense Baselines
>
>
> We again thank the reviewer for following up on the external defense baselines. Following the new pointers from the reviewer, we ran new evaluations to compare to the SmoothLLM and Self-Reminders defense baselines.
>
> | Method                             | ASR on the WildJailbreak Adversarial Evaluation Set (2K+ examples) $\downarrow$ |
> |------------------------------------|--------------------------------------------------------------------|
> | Tulu2                              | 71.0                                                               |
> | Tulu2 (+ trained on WildJailbreak) | 1.7                                                           |
> | Tulu2 (+ Self-Reminders)           | 41.9                                                               |
> | Tulu2 (+ SmoothLLM)            | 42.2                                                               |
>
> Our results show that applying the two external defense methods to Tulu2 reduces the attack success rate (ASR) from 71% to around 42%. However, training Tulu2 on WildJailbreak decreases the ASR to just 1.7%, making it significantly more effective than Self-Reminders and SmoothLLM. Additionally, it’s worth noting that the SmoothLLM defense requires duplicating and perturbing each prompt many times (in our experiment, five), which substantially slows down model inference. We hope these experimental results further strengthen the reviewer's confidence in the effectiveness of our safety alignment dataset.
>
> Lastly, we would like to take a moment to clarify why we consider extrinsic defense comparisons tangential to achieving our primary research goal of training safer models in our initial rebuttal. While we agree that model defense requires systematic solutions at various intrinsic and extrinsic stages, our work focuses specifically on *training a model to be safer by developing an improved safety training resource*. Extrinsic defenses address challenges very different from making a model intrinsically robust. The main issue is that applying extrinsic defenses to mitigate real-world harms is impractical with the current usage of *open-source* language models. Users typically interact with these models by inputting queries directly into the model and receiving outputs, but there's no way to enforce the use of extrinsic defenses during this process—like prompt engineering (Self-Reminders), perturbation injections (SmoothLLM), or safety filters (LlamaGuard)—on open-source models. These methods rely on voluntary application, which malicious users would avoid when exploiting models for harmful purposes. As a result, extrinsic defenses cannot replace intrinsic safeguards within the models themselves, making developing safety training resources, like WildJailbreak, essential and an independent challenge.
>
> *We will add the results of the new defense baselines and the discussion of internal and external defense to the final paper to further ground our contributions. We greatly appreciate the reviewer's raising this point to help us enrich the discussion!*

---

> > ### Comment · Reviewer_ZrdC · 2024-08-12
> >
> > Thanks very much for your detailed and insightful responses. All my concerns have been addressed now. I would raise my score to 7 to recommend an acceptance.

---

> ### Author Response · Authors · 2024-08-12
>
> Dear Reviewer `ZrdC`,
>
> Thank you very much for your constructive feedback throughout the review, rebuttal, and discussion period. We're pleased that our discussions helped address your concerns, and we're thrilled to see the improved score. We will ensure that our discussions are fully incorporated into the final paper.
>
> Sincerely,
> The Authors of Paper 12439

---

### Official Review · Reviewer_e44H · 2024-07-11

**Soundness:** 3
**Presentation:** 3
**Contribution:** 3
**Rating:** 5
**Confidence:** 4

**Summary:**

This paper introduces WILDTEAMING, an automated red-teaming framework designed to enhance the safety of large language models (LLMs) by identifying and mitigating jailbreak tactics from user interactions. The framework consists of two main steps: mining in-the-wild user-chatbot interactions to discover novel jailbreak tactics and composing these tactics into more challenging adversarial attacks. The paper also presents WILDJAILBREAK, a large-scale open-source safety dataset generated through WILDTEAMING, containing 262K prompt-response pairs designed to improve the safety and robustness of LLMs.

**Strengths:**

1. WILDTEAMING leverages real-world user interactions to identify and compose jailbreak tactics, providing a more diverse and realistic set of adversarial challenges than previous methods.

2. The creation of WILDJAILBREAK as an open-source resource is a significant contribution, offering a comprehensive dataset for safety training that addresses both harmful and benign queries.

3. The paper provides thorough evaluations of the framework's effectiveness, including comparisons with other state-of-the-art jailbreak methods and detailed analyses of the safety improvements achieved.

**Weaknesses:**

1. While the evaluation is comprehensive, it primarily focuses on specific benchmarks and models. Broader evaluations across different types of LLMs and use cases might be necessary to generalize the findings.

2. The framework heavily relies on existing datasets and moderation tools, which may have inherent biases or limitations that could affect the results.

**Questions:**

1. How well does WILDTEAMING generalize to LLMs beyond those tested in the paper? Have there been tests on a wider variety of models, including those with different architectures or training data?

2. How does the robustness of models trained with WILDJAILBREAK hold up over time as new adversarial tactics emerge? Are there mechanisms in place to continuously update the dataset and training methods?

3. What specific measures will be taken to prevent the misuse of the jailbreak tactics and the WILDJAILBREAK dataset?

**Limitations:**

1. The effectiveness of WILDTEAMING and WILDJAILBREAK might be influenced by the specific models and datasets used in the experiments, potentially limiting the generalizability of the results.

2. The framework's reliance on large-scale data mining and processing could be resource-intensive, potentially limiting its accessibility and applicability for smaller organizations or individual researchers.

---

> ### Author Rebuttal · Authors · 2024-08-06
>
> We greatly appreciate reviewer `e44H` for recognizing our work’s unique contribution in identifying a “more diverse and realistic set of adversarial challenges than previous methods”, our significant open-source contribution, and our “thorough evaluations” and “detailed analyses”. **We respond to four shared comments in the general response**, including:
> - Ethical Considerations
> - Scalability of WildTeaming and Analysis of Computation Cost
> - Dependency on Existing Building Blocks & Method Generalizability
> - Long-Term and Evolving Impact
>
> We will address other individual questions in the following section and are happy to follow up during the discussion period for any further inquiries.
>
> > Accessibility of WildTeaming
>
> In the general response, we have partially addressed reviewer `e44H`’s concern about WildTeaming's accessibility by showing that the method is scalable with reasonable computational cost. For details, please refer to the “Scalability of WildTeaming and Analysis of Computation Cost” section of the general response.
>
> To further address reviewer `e44H`’s concern that WildTeaming may rely heavily on large-scale data mining and may be prohibitive for researchers with fewer resources, we want to assure the reviewer that we have taken steps to mitigate this potential issue. Firstly, the initial mining of in-the-wild jailbreak tactics is a one-time effort that we have already completed. Researchers can reuse our publicly released jailbreak tactics to create new attacks as needed flexibly. Composing adversarial attacks with these selected jailbreak tactics requires only inference-time computational resources, which are typically inexpensive. Most importantly, to further reduce computational burdens, we have openly released the large-scale WildJailbreak dataset, making it a go-to safety alignment resource fully accessible for anyone to use and adapt.
>
> > Generalizability of Evaluation Results
>
> In response to reviewer `e44H`'s question about the generalizability of our results, we acknowledge that the choice of evaluation tasks and models can influence generalizability if narrowly selected. However, we believe our paper presents a comprehensive evaluation that supports making generalizable conclusions. Due to space constraints, we included only the most representative results in the main paper. **The full results, which exhibit similar trends, are provided in the Appendix to save space and avoid redundancy in the main text.** For instance, in the jailbreak experiments shown in Table 2, we selected three representative white-box and black-box models of various sizes—Vicuna-7B, Mixtral (8×7B), and GPT-4. We also evaluated other models, including Tulu2 DPO (7B), Mistral (7B), and GPT-3.5, as reported in Table 18 of the Appendix. The evaluation results of all these models demonstrate the same trend and consistently support our conclusion that WildTeaming offers substantial advantages over other jailbreak baselines in identifying diverse successful attacks.
>
> Similarly, for the safety training experiments reported in Table 4, we only had space to present results for two general capability tasks, three vanilla safety capability tasks, and three adversarial safety capability tasks. However, we considered and evaluated many more tasks across many ablation experiments, with full results provided in Tables 31, 32, 33, 34, 35 in §D.4 of the Appendix. Detailed descriptions of the complete suite of evaluation tasks is elaborated in §D.2 of the Appendix. Here's a summary of all tasks.
>
> #### Our *general capability* evaluation consists of **nine** benchmarks:
>
> - MMLU (language understanding)
> - GSM8K (math)
> - BBH (challenging multiple-choice questions)
> - TydiQA (multi-lingual multiple-choice questions)
> - CodexEval (programming problems)
> - TruthfulQA (truthfulness)
> - AlpacaEval 1 (chat ability)
> - AlpacaEval 2 (chat ability)
> - MT-Bench (multi-turn conversations).
>
> #### Our *vanilla safety* evaluation consists of **three** main benchmarks with fine-grained subcategory scores:
>
> - HarmBench (safeguard regarding standard, contextual, and copyright harmful behaviors)
> - ToxiGen (toxic and benign statements about 13 minority groups)
> - XSTest (safeguard regarding vanilla harmful queries; over-refusal against vanilla benign queries)
>
> #### Our *adversarial safety* evaluation consists of **three** main benchmarks with fine-grained subcategory scores:
>
> - JailbreakTrigger (13 categories of jailbreak attacks identified in previous work)
> - Do-Anything-Now (templated human-written jailbreak prompts)
> - WildJailbreak (safeguard regarding adversarial harmful queries; over-refusal against adversarial benign queries)
>
> **Having thoroughly evaluated safety-trained models across 15 benchmarks, with even more fine-grained scores on subtasks, we can confidently conclude that WildJailbreak is a high-quality, safe alignment dataset.** This dataset supports models in achieving balanced safety behaviors: providing appropriate safeguarding without over-refusal, effectively handling both vanilla and adversarial queries, and maintaining general capabilities with minimal, if any, decrease.
>
> **Finally, we follow rigorous standards in experiment design to avoid potential train-test contamination risks.** Our method development and dataset creation were independent of the selected evaluation benchmarks, metrics, and classifiers. We also utilized different safety classifiers within the attack composing stage of WildTeaming than in downstream tasks to avoid over-optimization of the evaluation metrics. We also ensured during safety training experiments, the train and test sets contained mutually exclusive data sets to prevent data contamination. **Thus, we are confident our evaluation is robust and provides strong evidence of WildTeaming's effectiveness in red-teaming and safety enhancement.**

---

### Author Rebuttal · Authors · 2024-08-06

We thank all reviewers for their positive reviews and constructive suggestions! We are enlightened to see all reviewers recognize our thorough experiments and evaluations. Additionally, we thank reviewers for recognizing our open-sourced effort (`e44H`, `ZrdC`, `idQi`), the novelty of our method (`e44H`, `ZrdC`, `idQi`), and the practicality of our work in improving LM safety (`idQi`, `wDdc`).

**Here, we will address shared comments from reviewers `e44H` and `idQi`, and address other comments in individual sections.** We’re more than happy to address any follow-up questions during the discussion period.

> Ethical Considerations

Due to initial page limits, we left the full ethics statement in §10 of the supplementary checklist. We will add an extended ethical discussion to the final paper as outlined below.

- **Dual Use Risks:** While WildTeaming aims to improve model safety, malicious actors could misuse our attacks to elicit harmful content from LM. Yet, the dataset does not introduce new harm but rather exposes existing vulnerabilities, allowing us to address them. Ignoring these loopholes won’t make models more robust, as malicious actors can already exploit models. Without resources like ours, these risks would persist to cause more long-term harm.

- **Strategies for Mitigating Misuses:** (1) We restrict access by requiring users to agree to a content warning and terms of service, ensuring valid justification to gain access. (2) We maintain records of contact info for accountability and proper usage. (3) Our resource release undergoes strict legal reviews by our institute's legal and ethics board.

- **User Privacy:** Our method is built on open-source LMSYS and WildChat datasets, which protect user information by not requiring account creation, hashing IP addresses, removing PII info, and strict internal legal reviews. Thus, our data inherits these protections. Moreover, our use of user data is limited to extracting high-level jailbreak tactics abstracted away from the concrete user content.

> Scalability of WildTeaming and Analysis of Computation Cost

WildTeaming's scalability is driven by two key aspects: (1) *Efficiency*: it is model-agnostic and optimization-free, and (2) *Diversity*: it combines various jailbreak tactics for diverse attacks. We emphasize that diversity is central to WildTeaming's scalability. Unlike methods that identify a narrow range of attacks, WildTeaming's diversity enables extensive, large-scale red-teaming in a single setting, broadly uncovering model vulnerabilities for improved robustness. Due to space constraint, we can share detailed explanations during discussion period.

Let's quantitatively compare the runtime and computational resources required for WildTeaming and other baselines in our paper, using NVIDIA RTX A6000 GPUs and Tulu2 DPO 7B as the target model.

| Method | Runtime (Sec) | GPU Require.                                  | Access Require. |
|------------|-------------------|----------------------------------------------------|---------------------|
| GCG        |             ~2400 | 1 for target                                     | White-box           |
| AutoDAN    |             12.55 | 1 for mutator; 1 for target                     | White-box           |
| PAIR       |             28.55 | 4 for attacker; 1 for target; GPT-4 for judger | White- & Black-box  |
| Ours        |              8.54 | 4 for attacker; 1 for pruner                    | White- & Black-box  |

WildTeaming generates an attack in 8.54s, the fastest among all methods. While WildTeaming and PAIR might appear to need more computational resources, the listed resources are based on our setup but can be reduced with 3rd-party hosting services like Together AI, suggesting our method can operate on lighter infrastructure, making direct resource comparisons less meaningful. Finally, WildTeaming attacks can be reused across multiple models, reducing the average computational overhead when red-teaming many models.

To address reviewer `idQi`'s concern about the scalability of WildTeaming with evolving models, we emphasize that WildTeaming is model-agnostic and not influenced by specific model sizes or characteristics. Our experiments (Figure 4) show that larger models do not consistently offer stronger defenses, indicating that WildJailbreak could remain challenging as models grow larger and more capable.

> Dependency on Existing Building Blocks & Method Generalizability

While the performance of WildTeaming is impacted by its building blocks, e.g., source data, attacker models & moderation tools, such reliance on existing components is common for AI/ML systems. These imperfect building blocks are used synergistically to achieve our goal—creating large-scale, high-quality safety training resources—thereby should not be seen as a drawback. Rather, the modular design of WildTeaming allows it to accommodate new components like updated data, stronger models & moderation tools over time, showing long-term potential. Our final paper will address related limitations and discussions to ensure full transparency.

>Long-Term and Evolving Impact

WildTeaming is a generalizable framework, adaptable to emerging user interactions, rather than a one-time effort. We expect future models to gradually adopt our data and address targeted vulnerabilities. **This improvement already represents a substantial advancement to the current state of AI safety.** Enhancing system robustness is an ongoing process due to the constant emergence of new attack tactics. WildTeaming can be repeatedly applied to uncover these tactics, improving model safety. Future work includes updating jailbreak tactics, developing more challenging attacks, and studying user-invented attacks for insights into evolving behaviors. Additionally, releasing our codebase allows others to explore and create challenging attacks. These points will be discussed in the limitations and future work section of the final paper.

---

### Comment · Area_Chair_M6nq · 2024-08-11
**Dear reviewers, please read and respond to authors' rebuttal (if you haven't done so)**

Dear reviewers, please read and respond to authors' rebuttal (if you haven't done so). Thanks.

Your AC

---

### Decision · Program_Chairs · 2024-09-25

**Decision:**

Accept (poster)

**Comment:**

This submission received four ratings (5, 7, 5 and 6), averaging 5.75, which is above the acceptance. During rebuttal, two reviewers actively interacted with the authors and the considerable effort by the authors well addressed their concerns. The other reviewers have not responded the authors' rebuttal. But after carefully reading these two reviewers' concerns and the summarization by the authors for the feedback, it seems that the authors provided a sufficient demonstration about the experiments, discussion and clarity. Thus, I recommend the acceptance and hope the authors carefully follow the reviewers' advice to improve the final version.